# The effector-triggered immunity landscape of tomato against *Pseudomonas syringae*

Fabien Lonjon[1,3], Yan Lai[1,3], Nasrin Askari[1], Niharikaa Aiyar[1], Cedoljub Bundalovic-Torma[1], Bradley Laflamme[1], Pauline W. Wang[1,2], Darrell Desveaux [1,2,4] ✉ & David S. Guttman [1,2,4] ✉

Tomato (*Solanum lycopersicum*) is one of the world's most important food crops, and as such, its production needs to be protected from infectious diseases that can significantly reduce yield and quality. Here, we survey the effector-triggered immunity (ETI) landscape of tomato against the bacterial pathogen *Pseudomonas syringae*. We perform comprehensive ETI screens in five cultivated tomato varieties and two wild relatives, as well as an immuno-diversity screen on a collection of 149 tomato varieties that includes both wild and cultivated varieties. The screens reveal a tomato ETI landscape that is more limited than what was previously found in the model plant *Arabidopsis thaliana*. We also demonstrate that ETI eliciting effectors can protect tomato against *P. syringae* infection when the effector is delivered by a non-virulent strain either prior to or simultaneously with a virulent strain. Overall, our findings provide a snapshot of the ETI landscape of tomatoes and demonstrate that ETI can be used as a biocontrol treatment to protect crop plants.

Tomato (*Solanum lycopersicum*) is the second most important vegetable crop (despite technically being a fruit) after potato, with a global production of approximately 190 million tons of fresh fruit in 2021 (United Nations FAOSTAT, fao.org). Tomato is a member of the Solanaceae family, which includes over 2700 species that show great diversity in their morphology and habitats, including potato, eggplant, tobacco, and petunia. The tomato clade includes cultivated tomato (*Solanum lycopersicum*) as well as twelve wild relatives that are endemic to South America[1]. Tomato plants are highly vulnerable to infectious diseases that manifest as wilts, leaf spots/blights, fruit spots, and rots[2]. Although fungal pathogens are most common, bacterial and viral infections also pose significant threats and regularly reduce crop yield and quality. For example, yield loss due to the phytopathogen bacterium *Pseudomonas syringae* can be as high as 75% in the Mediterranean basin, while in Ontario, bacterial diseases can cause yield losses up to 60%[2,3].

*Pseudomonas syringae* is a diverse bacterial species complex commonly found in soil, water, and on plant surfaces[4–7]. It is known to cause disease in a wide range of plant species, including crops, fruits, and ornamental plants. Due to its economic impact on agriculture, *P. syringae* has been extensively studied as a model pathogen for understanding plant-microbe interactions[4]. *Pseudomonas syringae* has a broad range of virulence factors, including toxins and effector proteins, which promote infection[4]; however, the most important virulence system is the type III secretion system and its associated type III secreted effectors (hereafter effectors), which are directly injected into host cells[7]. While the *P. syringae* species complex carries at least 70 distinct effector families, each individual strain harbors a unique repertoire of 10–50 effectors[8–10]. The primary function of effectors is to hijack plant defenses and promote bacterial disease development by suppressing plant immunity and creating a favorable growth environment[11–13]. However, plants have evolved intracellular immune receptor proteins (R proteins or Nucleotide-Binding Leucine-Rich Repeat Receptors, NLRs) to directly or indirectly detect pathogen effector proteins. The recognition of pathogen effectors by plant NLRs leads to a robust effector-triggered immune (ETI) response, which can be associated with a hypersensitive cell death response (HR)[14]. In tomato, the best characterized *P. syringae* ETI response is mediated by

[1]Department of Cell and Systems Biology, University of Toronto, Toronto, ON, Canada. [2]Centre for the Analysis of Genome Evolution & Function, University of Toronto, Toronto, ON, Canada. [3]These authors contributed equally: Fabien Lonjon, Yan Lai. [4]These authors jointly supervised this work: Darrell Desveaux, David S. Guttman. ✉e-mail: darrell.desveaux@utoronto.ca; david.guttman@utoronto.ca

the Pto-Prf protein complex[15–21]. The Pto kinase associates with the NLR protein Prf and interacts directly with the unrelated *P. syringae* effectors AvrPto and HopAB, which derepresses the Pto/Prf complex, leading to ETI. Pto was originally identified in a wild relative of tomato, *Solanum pimpinellifolium*, and it has been introgressed into many processing-type tomato varieties. The increasing incidence of *P. syringae* strains that can overcome Pto/Prf-mediated ETI in the field underscores the need to identify new sources of resistance[22].

In order to study the global *P. syringae* ETI landscape, we previously analyzed 494 *P. syringae* strains that represent the global effector diversity of the species complex and identified 529 effector alleles distributed among 70 distinct effector families[9,10]. These alleles were synthesized, cloned, and transformed into the *P. syringae* pv. *tomato* strain DC3000 (PtoDC3000, a tomato isolate that is also highly virulent on *A. thaliana*) to create the *P. syringae* type III effector compendium (PsyTEC)[23]. Systematic analyses of the ETI landscape of the model plant *A. thaliana* ecotype Col-0 against *P. syringae* identified 69 alleles from 21 families that can trigger ETI[23,24]. Almost all strains in the *P. syringae* species complex carry at least one ortholog of an ETI-eliciting effector allele, indicating that the presence of ETI is pervasive in this pathosystem[23]. A subsequent study of the oilseed crops *Brassica napus* (canola) and *Camelina sativa* (false flax) identified that 15 and 18 of the 19 *A. thaliana* ETI responses are conserved[25], suggesting that ETI is a prominent feature of Brassicaceous plants. However, it is yet to be confirmed if such a widespread ETI landscape is present in plant species outside the Brassicaceae family[26].

Plants have also developed diverse mechanisms to prime their innate immune system for more effective responses against subsequent biotic stresses[27]. Priming can be induced through microbial interactions in the rhizosphere or with foliar tissues, resulting in induced systemic resistance (ISR) and systemic acquired resistance (SAR), respectively[28]. These include ETI-inducing interactions, which have been shown to activate SAR[28]. In addition, priming can be induced by chemicals such as the SAR signaling molecules pipecolic acid and salicylic acid (SA), or the synthetic SA analogs 2,6-dichloroisonicotinic acid (INA) and benzothiadiazole (BTH)[29,30]. In tomatoes, both chemical and microbial priming have been shown to protect against various pathologies[31]. It has been demonstrated that pre-activation of ETI, utilizing an inducible effector expression system, can enhance priming against PtoDC3000 infection in *A. thaliana*[32]. However, the effectiveness of using microbially-induced ETI as a targeted immunostimulant for priming in tomatoes has not yet been established.

Building on the hypothesis that ETI can be used as an effective biocontrol agent in tomatoes, the objectives of this study were twofold. First, we sought to identify and characterize *P. syringae* effectors that can trigger a robust ETI in tomato plants. For this purpose, we developed a high-throughput ETI screen to survey PsyTEC for strong ETI-elicitors in several cultivated and wild tomato varieties. Second, we sought to use the identified ETI-elicitors to establish an immunoprotective biocontrol assay to protect tomatoes against disease. Overall, our screen identified six ETI-eliciting effector families in tomatoes including five families that were previously uncharacterized. Recognition of these six families is broadly distributed in both wild and cultivated tomato varieties. Finally, we demonstrate that these ETI responses can provide robust disease resistance when used as immunospecific priming agents.

## Results

### Divergence of the tomato and *A. thaliana* ETI landscapes
We first assessed the conservation of ETI responses between *A. thaliana* ecotype Col-0 and tomato cv. Glamour among the 21 families of *P. syringae* ETI-eliciting effectors previously identified in *A. thaliana* PsyTEC screens[23,24]. One ETI-eliciting effector allele was selected from each of the 21 families and transformed into the strain *P. syringae*

pv. *tomato* DC3000 (PtoDC3000)[23], which is highly virulent on both *A. thaliana* and tomato. We spray-inoculated three- to four-week-old Glamour plants and quantified bacterial growth four days post-infection. Our virulent control was PtoDC3000 carrying an empty vector (EV), and our ETI positive control was PtoDC3000::HopAB1n, which is known to elicit ETI in tomato (but not *A. thaliana*)[16]. We expected that ETI-eliciting effectors would reduce the *in planta* bacterial growth relative to the PtoDC3000::EV virulent control. Our growth assays found a remarkably low level of conservation between the ETI profiles in tomato and *A. thaliana*, with no tomato ETI response for 17 of 21 (81%) effector families that elicited ETI in *A. thaliana* (Fig. 1). Only three effectors, HopAA1q, HopAR1h, and HopBJ1b, significantly reduced PtoDC3000 growth relative to the EV control, while HopBA1a showed an inconsistent and weak ETI response, with only two of five replicate experiments showing significant reductions in growth. These data suggest that the ETI landscape of tomato against *P. syringae* is significantly different from *A. thaliana*.

### Comprehensive ETI screen of cultivated tomatoes
We developed a high-throughput screen to identify ETI responses elicited by the 402 expressed alleles from the PsyTEC library[23]. This screen prioritized throughput over sensitivity, and therefore, focused on identifying strong and consistent ETI responses. While weak or inconsistent ETI responses may have been missed with this approach, it did identify robust ETI responses that are more likely to be of value for crop protection.

We sprayed inoculated PtoDC3000 carrying individual PsyTEC effector alleles on three- to four-week-old tomato plants and quantified the outcome of each assay by measuring plant tissue damage (i.e., decline in green tissue) using the ImageJ macro PIDIQ (plant immunity and disease image-based quantification)[33]. The PIDIQ values were normalized relative to the ETI-eliciting (PtoDC3000::HopAB1n) and virulent (PtoDC3000:EV) controls, which were set to values of 0 and 100, respectively, resulting in a normalized disease score with low values corresponding to ETI outcomes and high values corresponding to disease outcomes. Screening of 402 alleles from 59 effector families showed that the distribution of disease scores generally fell into two groups, with a small number of alleles giving rise to low disease scores (0–30% decline in green), and a much larger number giving rise to high disease scores (60–100% decline in green) (Fig. 2a, b). Consequently, we used a disease score threshold of 40 to categorize the interaction outcomes into ETI-eliciting and non-ETI-eliciting (Fig. 2a). Using this threshold, we identified 23 alleles from six effector families as ETI-eliciting in tomatoes, including HopAB (7 alleles), HopAA (3 alleles), HopAR (6 alleles), and HopBJ (2 alleles). We also identified two previously uncharacterized families, HopBC (2 alleles) and HopBF (3 alleles), and confirmed these findings via bacterial growth assays (Fig. 2b; Supplementary Figs. 1 and 2; Supplementary Data 1). We did not identify any ETI-elicitors from the HopBA1a family, which was shown to be weak and inconsistent in the smaller, initial screen (Fig. 1). Further, we did not capture any intermediate ETI responses (30–60 % decline in green) as we did in *A. thaliana* (Fig. 2c)[23].

To explore cultivar variation in ETI responses, we expanded the PsyTEC screen to four additional tomato cultivars (Vendor, Cherokee Purple, SubArctic, and Moskvich). For this, we further increased the throughput of the screen by infecting ten-day-old seedlings grown in 50-well flats. Although the screens did not uncover any new ETI-eliciting alleles we did observe immunodiversity across the cultivars (Supplementary Figs. 3 and 4). HopAA and HopBC ETIs were conserved in Vendor, Cherokee Purple, and SubArctic, but completely lost in Moskvich. The HopAR ETI was only observed in Vendor, but only in two of the six alleles previously identified (HopAR1c and HopAR1e). In contrast, HopAB and HopBF ETIs were entirely conserved among all four cultivars.

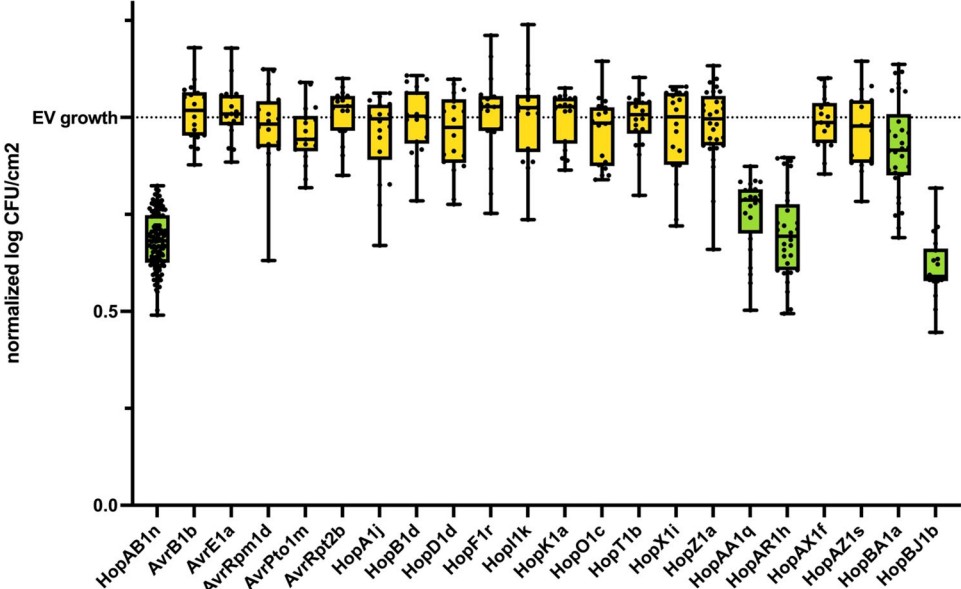

**Fig. 1 | Tomato ETI response of *P. syringae* alleles that elicit ETI on *A. thaliana*.** Growth of PtoDC3000 strains carrying one of 21 type III effectors known to elicit ETI on *A. thaliana* Col-0[23,24]. Growth data was normalized based on the growth of the virulent control PtoDC3000::Empty Vector (EV). PtoDC3000::HopAB1n served as a positive control for ETI elicitation. The green boxes indicate effectors that caused a significant reduction in bacterial growth compared to the EV (ANOVA *post-hoc* Tukey-test [2-tailed], $P < 0.05$). The box plots represent pooled data from four (HopBA1a, HopZ1a, HopAR1h and HopAA1q) or three independent experiments, each with six to 12 replicates ($n = 30$ for HopBA1a, HopZ1a and HopAR1h, $n = 24$ for HopAA1q, $n = 18$ for the rest of effectors). Error bars representing SEM. The dots show individual values, while the boxes display the first quartile, median, and third quartile, with whiskers extending to the smallest and largest values. Data represent the normalized results of twelve experiments. The minimum growth observed for the EV strain was 6.72 log CFU/cm², the maximum was 7.32 log CFU/cm², with an average of 7.12 log CFU/cm². For normalization within a single experiment, the growth of a particular replicate was divided by the average growth of the empty vector strain within that specific experiment.

## Comprehensive ETI screen of wild tomatoes

Wild relatives of tomatoes have long been recognized as a critical source of genetic diversity for crop breeding programs[34]. We first tested if the ETI responses identified in cultivated tomatoes were conserved in two wild species, *Solanum arcanum* TS713 and *Solanum pimpinellifolium* LA1547. We chose one representative allele from each ETI-eliciting family and transformed them into the PtoDC3000 lacking the effectors AvrPto and HopAB to avoid any possibility that these effectors would trigger ETI responses in our host. We observed strong ETI responses for HopAB1n, HopBJ1b, and HopBF1a in both species (Fig. 3a). The HopBC1b allele triggered an ETI in *S. arcanum* but not in *S. pimpinellifolium*. Finally, HopAR1h and HopAA1q ETIs, which were seen in cultivated tomatoes were lost in both species (Fig. 3).

We next comprehensively screened PsyTEC for ETI in the two wild tomato species using four-week-old seedlings (Fig. 3b, c; Supplementary Figs. 5 and 6) Table 1. The ETI conservation between *S. pimpinellifolium*, *S. arcanum*, and tomato cv. Glamour was very high, with no new ETI-eliciting families identified in the two wild species (Fig. 3b, c). However, two new HopAB alleles elicited ETI in *S. arcanum* (HopAB1d and HopAB1g); (Supplementary Fig. 7; Supplementary Data 2). HopBF (3 alleles) and HopBJ (2 alleles) ETIs were fully conserved in both *S. pimpinellifolium* and *S. arcanum*, whereas HopAA (4 alleles) and HopAR (6 alleles) were absent (Fig. 3b, c). HopBC ETIs (2 alleles) were conserved in *S. arcanum* but not in *S. pimpinellifolium*. The HopAB ETI responses were conserved in both species except for HopAB1v ETI, which was absent in *S. pimpinellifolium* (Fig. 3b, c; Supplementary Fig. 7; Supplementary Data 2).

## Immunodiversity of wild and cultivated tomato

We next investigated the immunodiversity of diverse tomato accessions to the ETI-eliciting effector families identified in our screen. We selected representative effectors from four families, HopAB1j, HopAA1q, HopBF1a, and HopBC1b, and screened a collection of 149 tomato accessions that included ten cultivated tomato accession of subspecies *Solanum lycopersicum* var. *lycopersicum* (SLL) from Mexico, 116 cherry tomato accessions of subspecies *Solanum lycopersicum* var. *cerasiforme* (SLC) from Mexico, Mesoamerica, Ecuador, and Peru, and 23 accessions of the currant tomato of species *Solanum pimpinellifolium* (SP), which is a wild species native to Peru and Ecuador. This collection was assembled to represent the genotypic and phenotypic diversity of tomatoes at their origin of domestication[35]. Since our screening method requires the host to be susceptible to the control (EV) strain, we first screened the collection with PtoDC3000$^{\Delta avrPto\Delta hopAB1}$::EV to identify and remove resistant accessions. This preliminary screen identified 18 resistant SLC accessions and two resistant SP accessions (Supplementary Fig. 8). Following this, three accessions (2 SLC and 1 SLL accession) among the 129 susceptible lines did not germinate for the next assays, resulting in 126 susceptible lines that we inoculated with the PtoDC3000$^{\Delta avrPto\Delta hopAB1}$ carrying one of the four selected ETI-eliciting effectors and qualitatively scored them for ETI based on disease symptoms (Supplementary Fig. 8). 21 accessions were resistant to all four elicitors, while 14 were susceptible to all four (Fig. 4a). Most of the lines were resistant to HopAB1j (68/108; 62.9%), with resistant lines found in both species and subspecies, particularly *S. pimpinellifolium* (13/17) and *S. lycopersicum* var. *lycopersicum* (9/9). Between 37.5% and 50% of the lines were resistant to HopAA1q, HopBC1b, and HopBF1a, with resistant lines found in all groups. Notably, most *S. pimpinellifolium* lines were susceptible to these three elicitors (respectively for HopAA, HopBC, and HopBF, 3/21, 1/17 and 1/20 resistant accessions) (Fig. 4b). Additionally, there is a high overlap pattern for the elicitors HopAA, HopBC, and HopBF in the SLC group, particularly in SLC Ecuador, with a similar response observed for 60% of the accessions. We assessed the overall pattern of resistance via principal components (PC) analysis based on each effector's vector of susceptibility and resistance across all accessions. Generally, HopAB1j is quite distinct from the other three effectors along both PC1 and PC2 (42.8% and 27.3% variance explained, respectively, Fig. 4c).

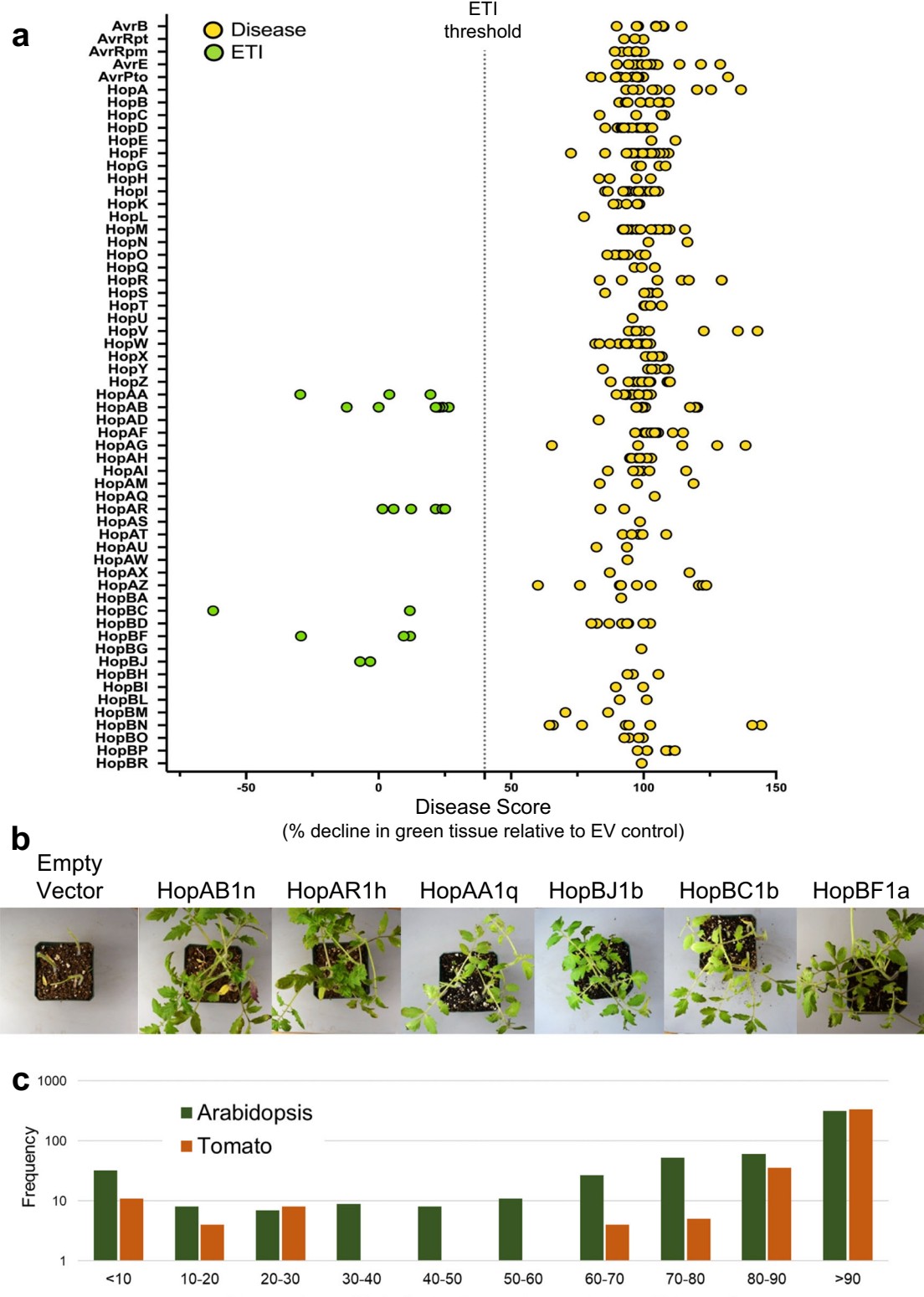

**Effector residues required for ETI and hypersensitive response elicitation**

Three of the six ETI-eliciting effector families have previously characterized enzymatic activities. HopAA is thought to contain a GTPase-activating protein (GAP) domain[36], HopAR is a well-characterized cysteine protease[37] while HopBF is a protein kinase[38]. We created strains with mutations in the putative catalytic sites of these three effectors, confirmed their expression (Supplementary Fig. 9a), and

then conducted bacterial growth assays and monitored disease symptoms following spray inoculations. Our results revealed that the catalytic residues of all three effectors are required to trigger ETI in tomatoes, although HopAA ETI was only partially lost in one out of three experiments (Supplementary Fig. 10).

Next, we determined whether the tomato ETI responses were associated with a hypersensitive response. We expressed a representative allele from each ETI-eliciting family and a catalytic mutant for

**Fig. 2 | The ETI landscape of *P. syringae* in tomato var. Glamour. a** Disease scores of PtoDC3000 carrying one of 402 expressed PsyTEC effector alleles were determined after spray inoculation on tomato plants. Disease scores were determined from the decline of green plant tissues throughout the experiment normalized to the negative control (PtoDC3000::EV, 100% decline in green) and positive control (ETI-eliciting allele PtoDC3000::HopAB1n, 0% decline in green). A cutoff of 40% (dashed line) was used to distinguish ETI-elicitors from non-ETI-elicitors. Yellow dots indicate alleles that do not elicit ETI, while green dots are alleles that elicit ETI. Each dot represents the average of at least two replicates (3–4 plants per replicate).

Raw data are presented in Supplementary Data 1 and Supplementary Fig. 1. **b** Representative pictures of tomato Glamour plants sprayed with PtoDC3000 expressing a representative allele of the six ETI-eliciting effector families or the empty vector. Pictures were taken 7 days post-inoculation and are from different experiments. **c** Distribution of disease scores (% green decline values) from the tomato var. Glamour PsyTEC screen (this study) (orange bars) and of the disease score values from the *A. thaliana* Col-0 PsyTEC screen[23] (green bars). The y-axis represents the number of effectors in each category when screened for ETI on the corresponding plant species and is presented as a log scale.

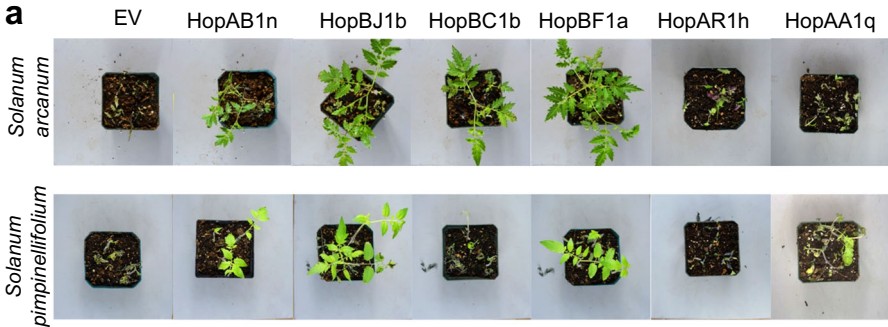

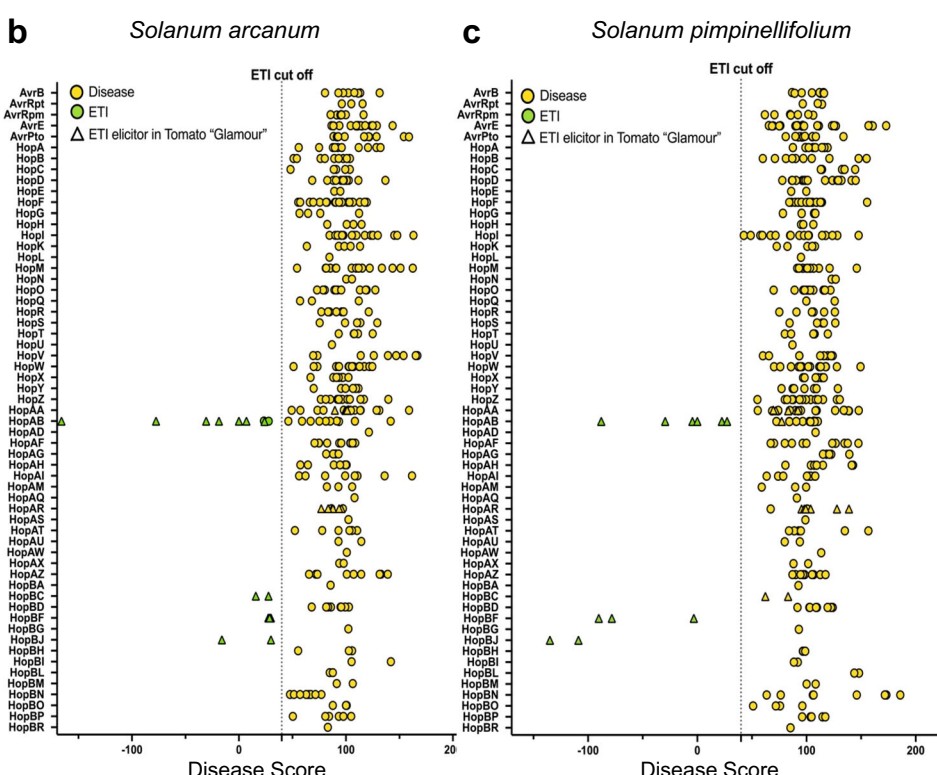

**Fig. 3 | The ETI landscape of *P. syringae* in wild tomatoes. a** Disease phenotypes of *S. arcanum* and *S. pimpinellifolium* plants after spraying with the PtoDC3000^ΔavrPtoΔhopAB1 expressing alleles of the five newly identified ETI-eliciting effector families and the empty vector (EV) disease and HopAB1n ETI controls. Symptoms pictured are 7 days post-inoculation. The pictures shown for the EV, HopAB1n, HopBJ1b, HopBC1b, and HopBF1 are from the same experiment. The pictures shown for HopAA1q come from a second experiment where appropriate EV and HopAB1n controls were also sprayed (see Supplementary Figs. 5 and 6). The

pictures shown for HopAR1h come from a third experiment where appropriate EV and HopAB1n controls were also sprayed (see Supplementary Figs. 5 and 6). Disease scores of PtoDC3000^ΔavrPtoΔhopAB1 carrying the 402 expressed PsyTEC effector alleles after spray inoculation on **b** *S. arcanum* and **c** *S. pimpinellifolium*. A cutoff of 40% (dashed line) was used to distinguish ETI-elicitors from non-ETI-elicitors. Yellow dots indicate alleles that do not elicit ETI; green dots are alleles that elicit ETI. Triangles indicate an ETI-elicitor in tomato var. Glamour. Each dot represents a pot of at least three plants.

**Table 1 | ETI-eliciting alleles**

| Effector family | N. Alleles[a] | ETI-eliciting, N (%) | ETI-Eliciting alleles | nRecPD[b] |
|---|---|---|---|---|
| HopAA1 | 20/1 | 3 (15.0) | i, q, t | 1.00 |
| HopAB1 | 19/5 | 7 (36.8) | j, n, p, q, r, v, ab | 0.86 |
| HopAR1 | 8/0 | 6 (75.0) | a, b, c, e, g, h | 0.30 |
| HopBC1 | 2/0 | 2 (100.0) | a, b | 0.50 |
| HopBF1 | 3/1 | 3 (100.0) | a, b, c | 0.64 |
| HopBJ1 | 2/0 | 2 (100.0) | a, b | 0.12 |

[a]Number of expressed alleles/Number of non-expressed alleles.
[b]nRecPD is an indicator of Horizontal gene transfer[41].

HopAA, HopAR, and HopBF in the effectorless strain PtoDC3000D36E[39] and confirmed the expression of each effector (Supplementary Fig. 9). We then infiltrated these strains into tomato cv. Glamour leaves and monitored tissue collapse 24 h post-inoculation. The empty vector strain did not trigger an HR, whereas the six ETI-eliciting alleles did (Supplementary Fig. 11). However, the ability to trigger an HR was abolished for the HopAR and HopBF catalytic mutants, whereas the HopAA catalytic mutant triggered an HR similar to that of the wild-type allele (Supplementary Fig. 11). Since HR has been associated with strong ETI responses, these results further support that our screen favored the identification of strong ETI responses[40].

We next assessed whether the newly identified ETI-eliciting families are recognized through the well-described Pto/Prf pathway. To this aim, we used the RioGrande PtoR, RioGrande PtoS, and Rio-Grande Prf3. RioGrande PtoR plants carry the PtoR gene and are resistant to *P. syringae* strains expressing AvrPto and HopAB[17], while RioGrande PtoS and RioGrande Prf3 carry a non-functional Pto locus and are susceptible (Supplementary Fig. 12). None of our newly identified ETI-eliciting effector families elicited ETI on RioGrande PtoR. Although we cannot rule out Pto or Prf dependency, we can conclude that the ETI requirements of HopAA, HopAR, HopBC, HopBJ, and HopBF are distinct from AvrPto and HopAB.

### Distribution of the tomato ETI load across the *P. syringae* species complex

When we previously analyzed the ETI landscape of *A. thaliana*, we found that 96.8% of the strains in the *P. syringae* species complex carried orthologs of ETI eliciting alleles, indicating that strains in the *P. syringae* complex face an extremely high potential ETI load when infecting *A. thaliana*. We performed the same analysis to assess how pervasive the *P. syringae* ETI load is when infecting tomatoes. We identified orthologs of all the effector alleles that elicit ETI on tomatoes and plotted their occurrence on the core genome phylogeny of 268 *P. syringae* strains selected as a non-redundant representative of the global species diversity (Fig. 5). In contrast to the *A. thaliana* results, only 39.9% (107 out of 268) of *P. syringae* strains harbor one or more orthologs of an ETI-eliciting effector, while 6.3% (17 out of 268) of strains harbor multiple ETI-eliciting effector orthologs. We also mapped the distribution of truncated ETI-eliciting alleles (i.e., those alleles that are less than 87% of the length of the known functional alleles) and found that 9% (24 out of 268) of the strains carry a putatively degenerate ETI-eliciting allele (Fig. 5; Table 2).

The ETI-eliciting families HopAR and HopBF are the most prevalent in the species complex (14.18% and 12.31% respectively), while HopBC and HopBJ alleles are found in only a small number of strains (4.58% and 2.24% respectively) (Table 2). Although the HopAA family is part of the conserved effector locus (CEL) and is found in most *P. syringae* isolates, only 4.5% of strains possess ETI-eliciting HopAA alleles (Table 2). When we compared the occurrence of ETI-eliciting alleles across the *P. syringae* phylogroups, we found that, except for phylogroup 3, more than 50% of strains in the primary phylogroups

carry at least one allele predicted to elicit ETI on tomato (Table 3). The vast majority of the primary phylogroup strains were isolated from agricultural plant hosts in contrast to the more divergent secondary phylogroup strains[10]. For example, in phylogroup 1, which includes the majority of tomato isolates, 63.5% of strains possess at least one allele orthologous to a tomato ETI-elicitor identified in this study (Table 3). This contrasts with strains in the secondary phylogroups, where only 14.2% of strains carry at least one ETI-eliciting effector.

We assessed the extent of horizontal transfer among the ETI-eliciting alleles via nRecPD, which compares individual gene genealogies to the core genome phylogeny to quantify the relative importance of non-vertical inheritance in the evolution history of a gene[41]. nRecPD values range from 0.0 to 1.0, with the former reflecting genes evolving primarily through the horizontal evolutionary process, and the latter reflecting genes evolving primarily through vertical descent[41]. We calculated the nRecPD values for the 23 tomato cv. Glamour ETI-eliciting alleles and found, similarly to previous observations in *A. thaliana*[8], that 13 alleles (56.5%) have nRecPD values near 1.0 and therefore have genealogies consistent with vertical inheritance (HopAA1t, HopAB1ab, HopAB1n, HopAB1p, HopAB1q, HopAB1r, HopAB1v, HopAR1c, HopAR1e, HopAR1h, HopBF1a, HopBF1b, and HopBJ1b). Alleles HopAA1q, HopBF1c, HopAB1j, HopBJ1a, and HopBC1a have nRecPD values between 0.5 and 0.8, and alleles HopAR1g, HopAR1b, HopAR1a, HopAA1i, and HopBC1b, exhibit low nRecPD values (<0.3), indicating a high degree of horizontal transfer among *P. syringae* isolates. When examined at the family level, these data recapitulate the negative association between the proportion of ETI-eliciting alleles in a family and the nRecPD value found in *A. thaliana* (i.e., families that largely evolve through vertical inheritance generally have a smaller proportion of ETI-eliciting alleles)[8], showing a Pearson $R^2 = 0.61$ association between family-level nRecPD and the proportion of ETI-eliciting alleles in each effector family when tested on tomato (Supplementary Fig. 13) Table 1.

### ETI protects tomatoes against concomitant or subsequent *P. syringae* infection

We investigated whether a *P. syringae* strain carrying an ETI-eliciting effector could protect tomatoes against infection by a virulent *P. syringae* strain. We co-infected tomato cv. Glamour plants with the avirulent effectorless polymutant PtoDC3000D36E[39] (D36E) carrying either HopBF1a or HopAB1n ETI-eliciting elicitors and the virulent parental strain PtoDC3000. To assess protection dynamics, we pre-infected the ETI-eliciting strain at 48 and 24 h prior to PtoDC3000 infection, infected both strains simultaneously, or post-infected the ETI-eliciting strain 24 and 48 h after PtoDC3000 infection. Disease symptoms were assessed seven days post-infection with PtoDC3000. Remarkably, we observed that co-inoculation or pre-treatment with D36E carrying HopBF1a or HopB1n at 24 h or 48 h pre-infection completely abolished disease symptoms caused by PtoDC3000 (Fig. 6a, b). No protection was observed when tomato plants were pretreated with D36E strain carrying an Empty Vector (D36E::EV), indicating that the protective effects against *P. syringae* infection are specific to the presence of HopBF1a or HopB1n elicitors. In contrast, when the D36E strain of *P. syringae* was inoculated after the virulent DC3000 strain, no protection against *P. syringae* infection was observed (Fig. 6b). This result suggests that the timing of effector application is critical for inducing effective resistance in tomato plants. In order to investigate the dose dependency of the ETI-induced protection against PtoDC3000, we conducted co-inoculation experiments with varying concentrations of D36E::HopAB1n (OD$_{600}$ 1.00, 0.33, 0.10, 0.033, and 0.01). Our results revealed that ETI-induced protection was dose-dependent. A robust protective effect was consistently observed against PtoDC3000 when D36E::HopAB1n was inoculated at the higher concentrations of OD$_{600}$ 1.00 and 0.33 (Supplementary Fig. 14a), whereas no significant protection was observed at the lower concentrations of 0.10, 0.01 and 0.033 (Supplementary Fig. 14a). To assess

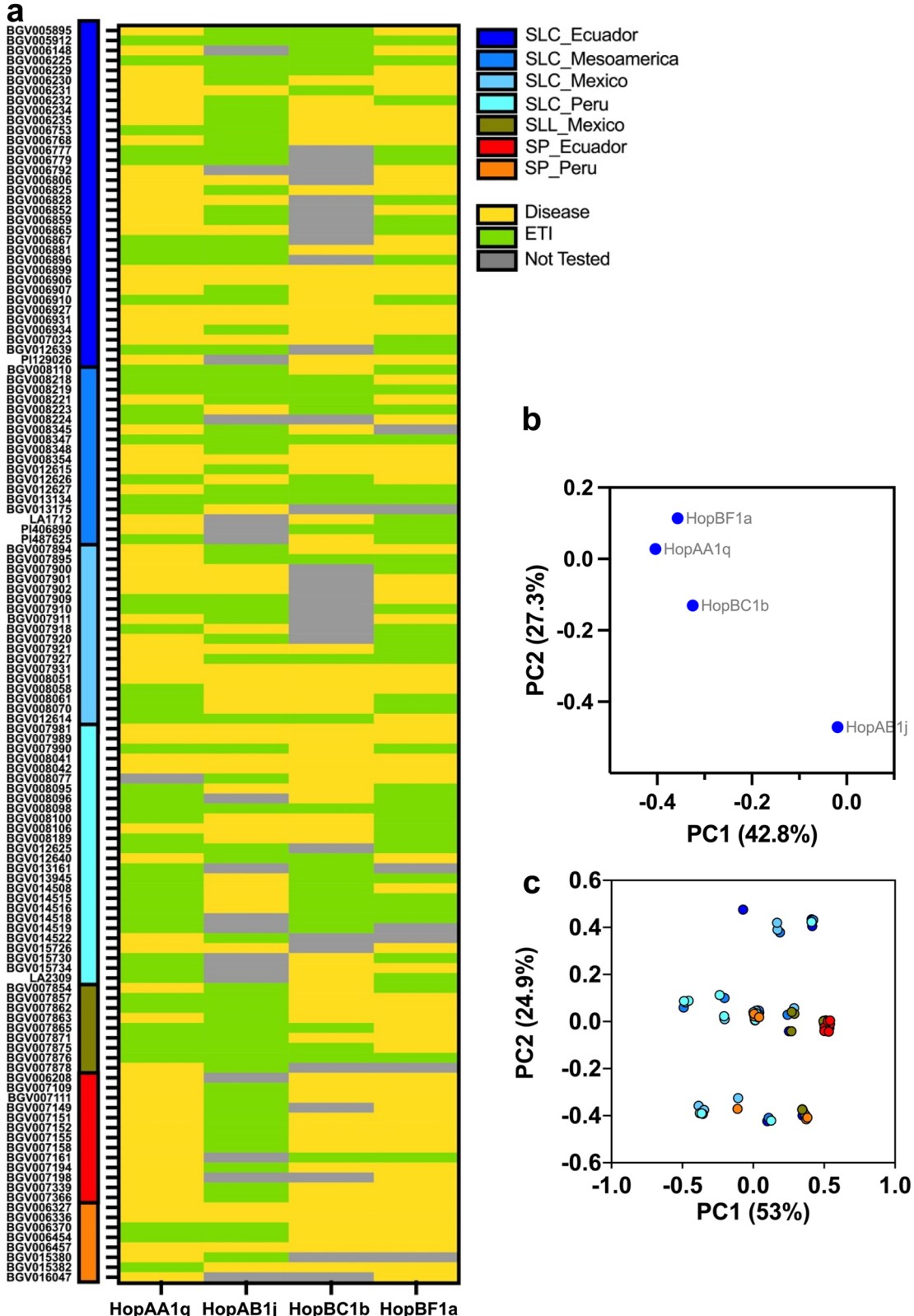

**Fig. 4 | Immunodiversity of wild and cultivated tomato. a** Heatmap representing the phenotypes of the 126 tomato lines[35] classified by species and origin on the y-axis sprayed by four different ETI-elicitors expressed in PtoDC3000$^{\Delta avrPto\Delta hopAB1}$ along the x-axis. Yellow indicates no ETI, and green represents ETI. A gray color indicates that the corresponding plant could not be tested because the relevant seed did not germinate. ETI was scored based on visual phenotypic data from

Figure S8. SP stands for *S. pimpinellifolium*, SLC represents *S. lycopersicum* var. *cerasiforme*, and SLL denotes *S. lycopersicum* var. *lycopersicum*. **b** Principal component analysis (PCA) based on the ETI response profile of the four tested effectors. **c** PCA of the ETI response profiles of the tomato species as indicated in the legend of (**a**).

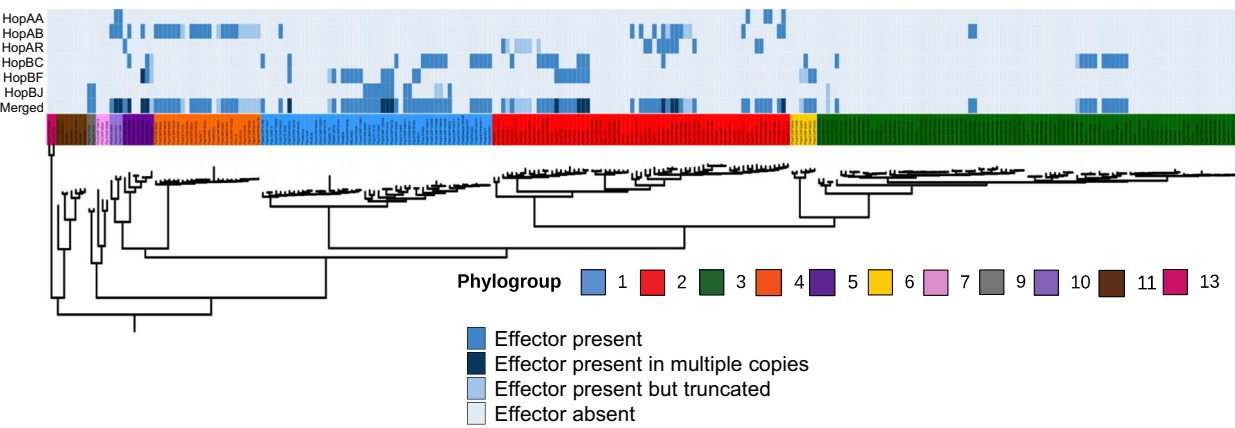

**Fig. 5 | The *P. syringae* ETI load on tomato.** Predicted orthologs of ETI-elicitors from tomato var. Glamour carried by each of the 268 non-redundant *P. syringae* strains used to generate the PsyTEC collection were mapped onto a previously generated *P. syringae* core genome phylogeny[23] with phylogroup designations. Effectors are color-coded as follows: darkest blue: effector present in multiple copies, dark blue: effector present in one copy, light blue: effector truncated, lightest blue: effector absent.

any potential fitness costs associated with D36E-induced ETI, we sprayed 4-week-old tomato plants with D36E::HopAB1n at $OD_{600}$ 1.00, 0.33, 0.10, 0.033, and 0.01 and measured the fresh and dry weight of the tomato plants 3 weeks post-inoculation. Our results indicate that there is no growth deficit associated with D36E::HopAB1n infection at any concentration tested, as the fresh and dry weights of inoculated plants were comparable to control plants sprayed with $MgSO_4$ (Supplementary Fig. 14b, c).

To provide further evidence that the observed protection is ETI-mediated, we pretreated RioGrande PtoR, PtoS, and Prf3, tomato plants with D36E::HopAB1n 24 h before PtoDC3000$^{\Delta avrPto \Delta hopAB1}$ infection. The results revealed that protection was observed only in the PtoR plants carrying a functional Pto locus, but not in PtoS and Prf3 plants, which lack a HopAB1n ETI response (Supplementary Fig. 15). These results demonstrate that the observed protection is ETI-mediated.

## Discussion

In this study, we identified novel sources of tomato resistance against *P. syringae* and have demonstrated that these resistance responses are broadly distributed across cultivated tomato species. In addition, we have demonstrated that ETI can be used as a robust priming agent to protect plants against infection.

We identified 25 alleles from six effector families that elicit ETI in tomatoes: 23 in cultivated tomato varieties and two specific to wild tomatoes. These include the well-characterized HopAB1 family as well as five families that had not previously been characterized as ETI-elicitors in tomatoes (HopAA, HopAR, HopBJ, HopBC, HopBF). This is the first report of HopBC and HopBF ETIs in any plant species.

**Table 2 | Frequency of ETI-eliciting alleles in the *P. syringae* species complex[a]**

| Effector family | One allele (%) | Two alleles (%) | Truncated alleles(s) (%)[b] |
|---|---|---|---|
| HopAA | 4.48 | 0.00 | 1.49 |
| HopAB | 8.58 | 0.37 | 1.49 |
| HopAR | 14.18 | 0.00 | 0.75 |
| HopBC | 4.48 | 0.00 | 2.24 |
| HopBF | 12.31 | 0.00 | 0.00 |
| HopBJ | 2.24 | 0.00 | 0.00 |
| Total | 33.6 | 6.3 | 9.0 |

[a]268 non-redundant strains used to generate PsyTEC[23].
[b]Truncations are defined as less than 87% of the length of the ETI-eliciting alleles.

The HopAB1 family is a well-characterized ETI-elicitor in tomatoes, and it was the most prominently recognized family in our study (Fig. 4). Interestingly, we observed that HopAB1j, native to PtoDC3000, elicits ETI in tomato plants when expressed on a multicopy plasmid but not when encoded on the chromosome, similar to what we previously observed with AvrE in *A. thaliana*[23]. We hypothesize that PtoDC3000 must carefully regulate the expression of these effectors to maintain their fitness advantages while avoiding ETI elicitation.

HopAA1 and HopAR1 are ETI-elicitors in *A. thaliana* with known NLR requirements. HopAR1, a cysteine protease, cleaves PBS1 and activates the NLR protein RPS5, leading to RPS5-dependent ETI[42]. While we found that HopAR1 cysteine protease activity is required to trigger ETI in tomato plants, the lack of RPS5 orthologs suggests that a different NLR is involved in its recognition[43]. HopAA1, which is part of the conserved effector locus (CEL), is present in most *P. syringae* strains[9,44] and triggers ETI via the NLR CAR1 in *A. thaliana*[23]. However, no ortholog of CAR1 is present in tomato, similar to the case of RPS5. This is perhaps not surprising since recognition of the same effector often involves different NLR proteins across plant species[45–48]. All three alleles of the HopBF1 family elicited ETI in tomatoes. HopBF1 is a kinase that phosphorylates the molecular chaperone HSP90[38]. HopBF1 kinase activity is required to trigger ETI, suggesting that a phosphorylated target, such as HSP90, may be NLR-guarded in tomatoes (Supplementary Fig. 10). HopBC1 and HopBJ1 are relatively rare effectors found in a minority of strains, with HopBC1 present in 4.48% of strains from phylogroups 2 and 5, and HopBJ1 present in 2.24% of strains from phylogroup 2 and 10. While HopBJ1 can also trigger ETI in *A. thaliana*, the associated NLR(s) remains unknown. Although the genetic requirements of the five novel tomato ETI-eliciting families remain to be determined, we know that their recognition mechanism is distinct from HopAB1 (Fig. 5).

The number of *P. syringae* strains encoding orthologs of tomato ETI-eliciting effectors was relatively low (39.9%) compared to *A. thaliana* (96.8%). Although tomato may in fact display a relatively limited ETI landscape against *P. syringae*, other factors may contribute to this difference. First, our high-throughput tomato screen favored throughput over sensitivity and, consequently, may not have captured the weak or intermediate responses that we were able to recover in *A. thaliana* (Fig. 2). For example, we did not identify relatively weak ETI responses such as those of HopBA1, which we observed only by growth assay (Fig. 1). Second, there is almost certainly additional, as yet unexplored tomato immunodiversity found in cultivars that we did not select for this study. We know this to be true since we did not identify AvrPto, a well-known ETI-elicitor in tomatoes. This is not a limitation of our

**Table 3 | Distribution of ETI-eliciting alleles in the _P. syringae_ species complex**

| Phylogroup[a] | N. Strains | % Strains with ETI-elicitor(s) | % Strains with truncated ETI-elicitor(s) |
|---|---|---|---|
| Phylogroup 1 | 52 | 63.5 | 5.8 |
| Phylogroup 2 | 67 | 50.7 | 13.4 |
| Phylogroup 3 | 95 | 13.7 | 2.1 |
| Phylogroup 4 | 24 | 62.5 | 29.2 |
| Phylogroup 5 | 7 | 57.1 | 0.0 |
| Phylogroup 6 | 6 | 50.0 | 33.3 |
| Phylogroup 7 | 3 | 0.0 | 0.0 |
| Phylogroup 9 | 2 | 100.0 | 0.0 |
| Phylogroup10 | 3 | 100.0 | 0.0 |
| Phylogroup 11 | 7 | 0.0 | 0.0 |
| Phylogroup 13 | 2 | 0.0 | 0.0 |
| Total | 268 | 39.9 | 9.0 |

[a]Phylogroups 1,2,3,4,5,6,10 are classified as primary phylogroups based largely on their more recent common ancestry[10].

screening method since we observed AvrPto ETI in the RioGrande cultivar (Fig. 5). Finally, it is possible that the PtoDC3000 strain used in our screen, which was isolated from tomato, may possess effectors that are effective at suppressing ETI on this host plant. However, the same strain was used for the _A. thaliana_ PsyTEC screen, which revealed pervasive ETI. Further, when we switched the screening strain to the Brassicaceous pathogen PmaES4326 we actually identified one more ETI response in _A. thaliana_ than PtoDC3000 (60 vs. 59). As such, we believe that tomato-specific ETI suppression is an unlikely explanation for the moderate ETI landscape of tomato. It is possible that tomatoes have a shorter co-evolutionary history with _P. syringae_ than _A. thaliana_, resulting in a more limited resistance profile against this pathogen. Probing the diversity of _P. syringae_ species in the Andes Mountains, where tomato originated, relative to areas of domestication may shed light on the evolutionary pressures that have shaped the current ETI landscape of tomato[1,49].

When we compare the specific patterns of ETI in tomato vs. _A. thaliana_ among the four effector families with more than three alleles that elicit ETI in tomatoes we find a variety of patterns (Fig. 7). All three HopAA alleles that trigger ETI in tomato var. Glamour are also recognized in _A. thaliana_, with one additional allele recognized in _A. thaliana_ (HopAA1m). Similarly, among the eight alleles of the HopAR family, both plants recognize six alleles, with an overlap of four alleles. In contrast, _A. thaliana_ does not recognize any of the 19 expressed HopAB alleles, while seven alleles elicit ETI in tomatoes.

Studies have demonstrated that domesticated plant crops are often more susceptible to pathogens than their wild relatives, and pathogens have evolved higher levels of virulence on domesticated hosts[50,51]. Additionally, domestication may reduce genetic diversity, decreasing the availability of resistance genes against pathogens and generally lowering immunodiversity[52,53]. As such, we hypothesized that wild tomato varieties might possess a higher immunodiversity than cultivated varieties. To investigate this, we screened PsyTEC on two wild tomato species, _S. arcanum,_ and _S. pimpinellifolium_, that are among the closest relatives of domesticated tomato and have the advantage of being self-compatible[54]. We were surprised to find a high degree of ETI conservation between the domesticated and wild tomato species, with four of six ETI-eliciting effector families conserved in _S. arcanum_, three of six conserved in _S. pimpinellifolium_, and no additional ETI eliciting families uncovered in the wild species. This would suggest that domestication has not significantly compromised the ETI potential of tomato, similar to what we previously observed in Brassicaceous plants[22]. In tomatoes, this might be explained by the high degree of NLR gene conservation between cultivated and wild-relative accessions[55].

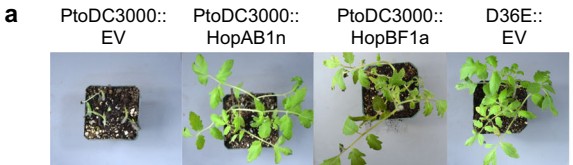

PtoDC3000:: EV   PtoDC3000:: HopAB1n   PtoDC3000:: HopBF1a   D36E:: EV

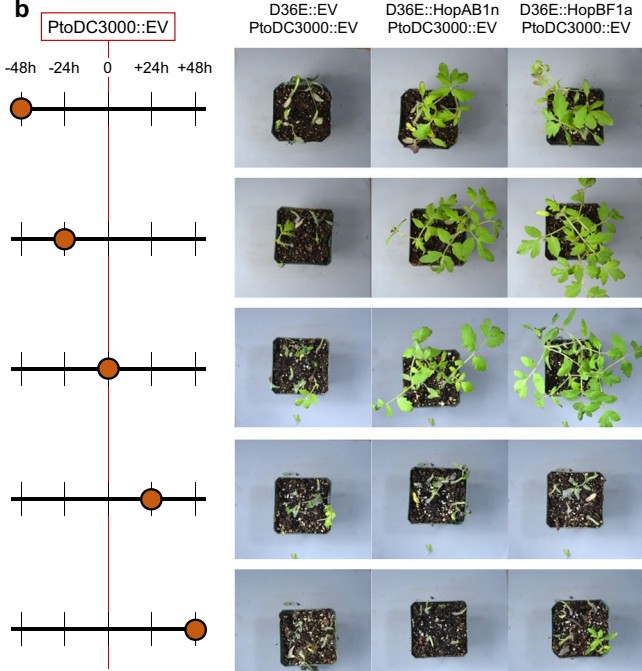

**Fig. 6 | ETI protects tomato plants against _P. syringae_ co-infection. a** Control infections with PtoDC3000 carrying an empty vector (EV), PtoDC3000 expressing the ETI-elicitors HopAB1n or HopBF1a, and the effectorless polymutant PtoDC3000D36E (D36E) strain[39] with the EV. Images were taken seven days post-PtoDC3000 inoculation. **b** Tomato var. Glamour plants were inoculated with the D36E strain at various time points relative to PtoDC3000 infection. The chart on the left indicates the relative infection time for each strain. PtoDC3000::EV was always infected at relative time 0, while D36E carrying either EV or an ETI eliciting effector was infected (orange circle) 48 or 24 h prior, simultaneously, or 24 and 48 h post-PtoDC3000::EV infection. Images were taken seven days post-PtoDC3000 inoculation. The experiments were repeated at least twice with similar results. The pictures shown for $t-48$ h and $t$-24h come from the same experiment. Picture for t0, $t+24$ h and $t+48$ h, DC3000::EV, HopAB1n, and D36E::EV come from another experiment. The picture for DC3000::HopBF1a comes from a third experiment. Pictures were taken 7 days post-inoculation.

Screening a collection of 126 tomato varieties (both wild and cultivated) with the four ETI-elicitors, HopAB1j, HopBC1b, HopAA1q, and HopBF1a, revealed a significant level of immunodiversity. The collection included the wild species _S. pimpinellifolium_, from which the _Pto_ locus was originally identified[56]. Most _S. pimpinellifolium_ lines tested were resistant to HopAB1j but were susceptible to HopBC1b, HopAA1q, and HopBF1a. This tomato collection has previously been used successfully to identify genes associated with phenotypic traits through GWAS[35]; however, attempts to map resistance loci for the four effectors screened were unsuccessful, possibly due to the size of the sample or the multigenic nature of the resistance in the population. Interestingly, 20 lines from the collection were resistant to PtoDC3000$^{\Delta avrPto\Delta hopAB1}$, supporting that there are still unidentified tomato ETI responses against _P. syringae_.

Finally, we demonstrated that ETI can be used as a priming agent in tomatoes to provide immunity against subsequent infection. Similar

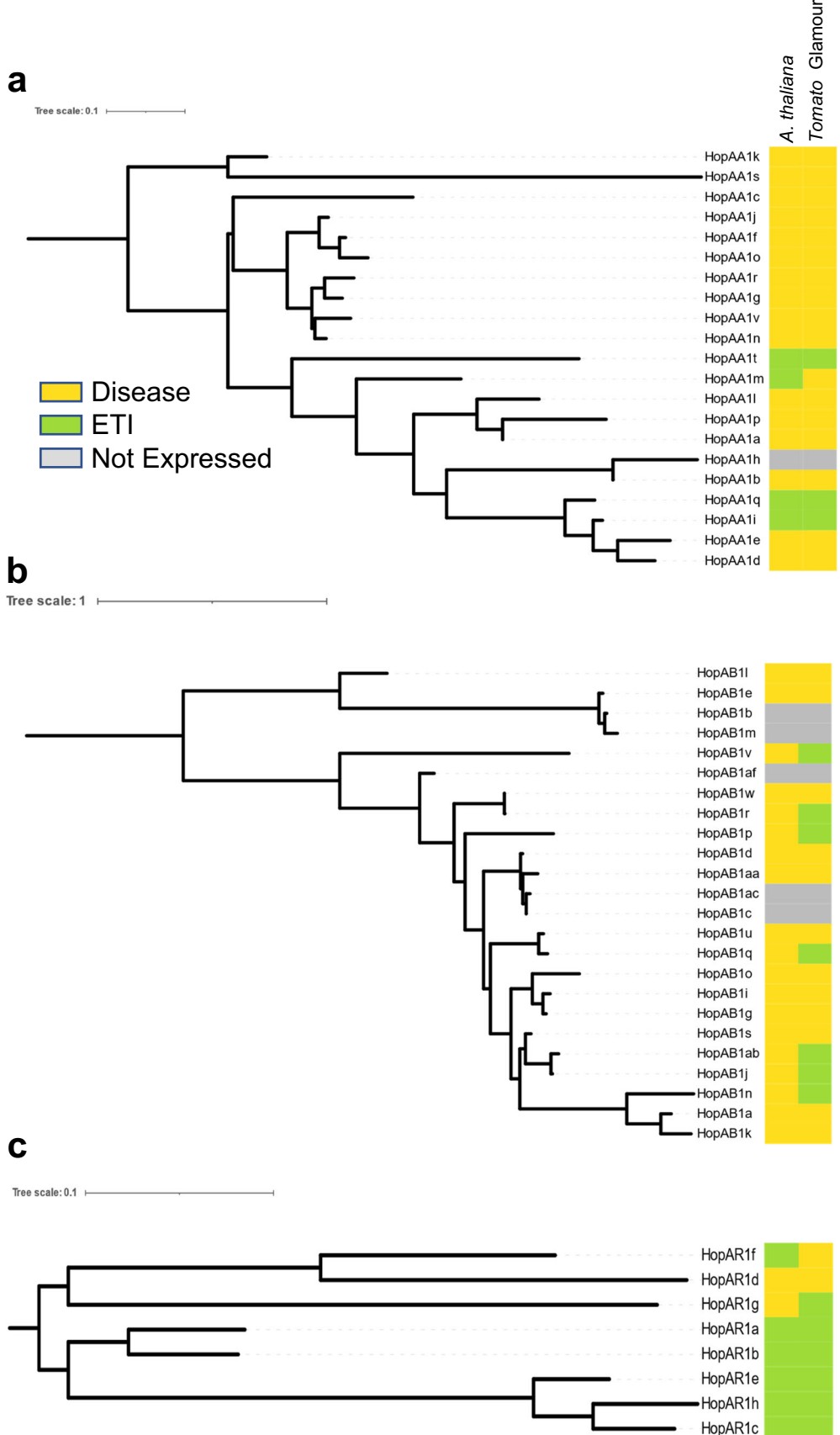

**Fig. 7 | Comparison of ETI responses in tomato var. Glamour and *A. thaliana*.**
Maximum likelihood phylogenetic tree for the PsyTEC-synthesized alleles for
**a** HopAA, **b** HopAB, and **c** HopAR. A yellow box indicates that the corresponding
allele does not trigger ETI, a green box indicates that the allele triggers ETI, and a
gray box indicates that the allele is not expressed in PtoDC3000 and could not be
tested.

to the systemic acquired resistance response (SAR)[27,28], this protection was found to be effective when the ETI-eliciting strain was pre-infected up to 48 h prior to infection with the virulent strains (Fig. 7). But in contrast to SAR, this ETI protection was also effective when the biocontrol strain was co-inoculated with the pathogenic strain (Fig. 7), which is, by definition, not a "priming" response, and may result from ETI acting as a 'dominant bad' in the population as we have previously observed in mixed population experiments[57]. As such, the resistance afforded by ETI protection may result from a combination of SAR-like priming as well as an effector-mediated communal detriment to the microbial community. However, inoculating the biocontrol strain after the pathogenic strain did not confer protection, indicating that ETI is not able to suppress the growth of a previously established virulent strain. This finding may be due to spatial ETI structure described as localized acquired resistance[58], where the ETI response occurs in a 2 mm area surrounding cells in contact with the elicitor[58]. This spatial structure implies that the primary protective impact of ETI may be localized to the initial infection site. Furthermore, it is possible that ETI-mediated protection involves the Pattern-Triggered Immune (PTI) response since these two branches of the plant immune system have been demonstrated to mutually potentiate one another[59–61].

Overall, the biocontrol approach taken in this study provides an exciting opportunity to rapidly deploy ETI against diseases without the requirement for breeding resistance genes into crops of interest. Mapping crop ETI landscapes against a broader range of pathogenic effectors can be used to guide the development of specific elicitor-based biocontrol that can be integrated into pest management practices and potentially reduce the reliance on chemical pesticides.

## Methods

### Bacterial strains and media

*Escherichia coli* strains were grown at 37 °C in Luria–Bertani medium. *P. syringae* strains were grown in KB agar (20 g/L peptone, 1.5 g/L $K_2HPO_4$, 15 g/L glycerol, and 15 g/L agar) supplemented with 3.2 mM $MgSO_4$. Effector expression was induced in *hrp*-inducing media (0.74 g/L $K_2HPO_4$, 6.226 g/L $KH_2PO_4$, 0.35 g/L $MgCl_2$, 0.099 g/L NaCl, 1.004 g/L $(NH_4)_2SO_4$, 0.5 M fructose, 0.2 M citric acid and 200 μM aspartic acid[23]). When needed, antibiotics were added at the following final concentrations (mg/L): kanamycin (50), rifampicin (40), tetracycline (10).

The PsyTEC (*P. syringae* Type III Effector Compendium) strains were generated in a previous study[23]. Tri-parental mating was used to generate each of the constructs into either the *P. syringae* PtoDC3000D36E[39] or the PtoDC3000 double mutant AvrPto and HopAB strain[62] as previously described[23,24].

Effector catalytic mutants were generated via PCR site-directed mutagenesis. A primer pair containing the desired mutations was used. PCR primers for HopBF1[D154A] were Fwd-'CCCACAACCTCCATATT GAGGCTTTACAGTTCATAATTGATGA'/Rev-'TCATCAATTATGAACTGT AAAGCCTCAATATGGAGGTTGTGGG' and for HopAR1[C98S] primers were Fwd-'CATAACAATATTAGTGCCGGCCTC'/Rev-'GAGGCCGGCACT AATATTGTTATG'. The HopAA1[LRA-FEN] mutant was previously described[20]. PCR products were treated with *DpnI* to digest the methylated parental DNA template. The resultant products were directly transformed into *E. coli*. The correct point mutations were verified by sequencing.

### Plant materials

Tomato lines used in that study are *S. lycopersicum* vars. Glamour (Stokes Seeds, Thorold, Canada), Vendor, Cherokee Purple, SubArctic and Moskvich (Vineland Research and Innovation Centre, Vineland Station, Canada) *S. pimpinellifolium* LA1547(TGRC, UC Davis, USA), *S. arcanum* TS713 (Gregory Martin, Cornell University, USA) and tomato RioGrande-PtoR, RioGrande-PtoS and RioGrande-Prf3[15]. The 146 tomato lines screened for ETI diversity (Fig. 4) were characterized in a

previous study[35]. Prior sowing, *S. pimpinellifolium* and *S. arcanum* seeds were soaked in a 2.7% sodium hypochlorite solution and rinsed 5 times with sterile water.

Tomato plants were grown in Sunshine Mix 1 soil at 24 °C during daylight and 22 °C at night, a light intensity of 150 umol/m2s, and a 12 h light / 12 h dark light schedule. Plants were sprayed with bacterial suspensions when they were three to four weeks old.

### Bacterial infection assays

Prior to spray inoculation, bacterial strains were grown overnight at 28 °C on KB agar supplemented with the appropriate antibiotics. Bacteria were resuspended in 10 mM $MgSO_4$ supplemented with 0.04% Silwet L-77 at a final $OD_{600} = 0.2$. 7.5 mL of bacterial suspension was sprayed onto a pot containing four 3–4 weeks old plants using Preval sprayers and immediately domed to maintain a high-humidity environment. For seedlings assays 10 days old plants were spray with 2 mL of bacterial suspension per cell. Plants were pictured 7 days post-inoculation using a Nikon D5200 DSLR camera affixed with a Nikon 18–140 mm DX VR lens. Quantification was performed using a PIDIQ ImageJ macro[33]. Bacterial growth assays were performed 4 days post-inoculation. Four leaf disks ($0.25 cm^2$ each), from the same tomato plant, were grinded and resuspended in 1 ml 10 mM $MgSO_4$. Bacteria counts were determined by plating dilutions on KB medium with appropriate antibiotics. Hypersensitive response assays were performed on 5-week-old plants. Plants were domed one day before infiltration. A bacterial suspension ($OD_{600} = 0.1$) was infiltrated into tomato leaves using a needleless syringe. Tissue collapse was monitored, and pictures were taken 24 h post infiltration.

### Validation of protein expression

Validation of protein expression for strains in this study was performed as previously described[23]. Effector expression was induced by inoculating strains to a final $OD_{600} = 0.1$ in *hrp*-inducing media and incubating overnight at 28 °C with shaking. 2 ml of culture was centrifuged and resuspended in SDS-loading buffer, followed by two 5-min incubations at 95 °C. These concentrated pellet samples were run on a 10% polyacrylamide gel and transferred to a nitrocellulose membrane for western blotting. Before immunoblotting membranes were soaked into a 5% glacial acetic acid and 1 g/L Ponceau S solution and rinse with distilled water to show equal loading. Antibodies used were anti-HA (Roche, 11867423001, 1:15,000) as a primary antibody and goat anti-rat antibody conjugated with horseradish peroxidase as the secondary antibody (Santa Cruz Biotechnology, sc-2032, 1:30,000). Signals were detected using Enhanced chemiluminescence (ECL) Western Blotting Substrates (Amersham, RPN2235).

### Phylogenetic tree

The strain collection was based on the 494 *P. syringae* isolate assemblies previously published by our group[10]. A reduced, non-clonal strain collection was identified as previously described[63]. In short, a single representative strain was selected from any pool of strains with a core genome evolutionary distance of less than 0.001 and a type III effector Jaccard similarity value ≥ 0.8. The core genome was identified with PIRATE (v1.0.4)[64], and annotated with Prokka (v1.14.6)[65]. IQTree (v1.6.12)[66] and the Ape (v5.7-1) R package[67] were used to generate core genome trees.

### Statistical information

Statistics in this paper were performed on the pool of all replicates for Fig. 1 and on individual replicates for Supplementary Figs. 2 and 10. All ANOVAs are two-way, using a *p*-value threshold for a significance of 0.05. Two-way *post-hoc* Tukey HSD tests were performed for all experiments. Significance groups were determined based on the Tukey HSD, using a *p*-value inference threshold of 0.05. Individual *p*-values are presented in Supplementary Data 3.

## Reporting summary

Further information on research design is available in the Nature Portfolio Reporting Summary linked to this article.

## Data availability

Data supporting the findings of this work are available within the paper and its Supplementary Information files. All vectors and resources described are available from the corresponding authors upon request. Source data are provided with this paper.

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

## Acknowledgements

We would like to thank Dr. Greg Martin for providing critical resources for the project, including TS713 seeds and the delta AvrPto HopAB strain, the C.M. Rick Tomato Genetics Resource Center (TGRC) for LA1547 seeds, the Vineland Research & Innovation Centre for providing tomato cultivars and Joaquin Cañizares Sales and Maria José Teresa De Jesus Diez Niclos for providing access to the Spanish accessions[35]. We would like to acknowledge all the members of the Guttman and Desveaux labs for their support and assistance, with particular mention to Dr. Lucía Graña-Miraglia for her assistance with the preparation of images. The Department of Cell & Systems Biology greenhouse staff of Bill Cole and Tom Gludovacz provided essential assistance during the bulking of the wild tomato accessions. Financial support was provided by George Weston Limited Seeding Food Innovation 2019 Program "Effector-Enabled Mining of Wild Plants for Novel Crop Immunodiversity" (SFI19-0354) to D.S.G. and D.D., and Mitacs Accelerate "Exploiting wild tomato genetic resources and pathogen effector diversity for resistance" (IT22231) to D.S.G., D.D., and F.L. Additional support was provided by National Science & Engineering Research Council of Canada Discovery Grants to D.S.G. and D.D.

## Author contributions

F.L., Y.L., D.D., and D.S.G. contributed to project conceptualization, writing, and editing. F.L., Y.L., Na.A., Ni.A., B.L., and C.B-T. contributed to data generation. F.L., Y.L., C.B-T, D.D., and D.S.G. contributed to figure preparation. P.W.W., D.D., and D.S.G. contributed to resource provision. D.D. and D.S.G. contributed to financial support and are listed alphabetically.

## Competing interests
