## [Peer Review File · Nature Communications]

The Effector-Triggered Immunity Landscape of Tomato against *Pseudomonas syringae*Reviewer #1 (Remarks to the Author):

In the paper entitled "The Effector-Triggered Immunity Landscape of Tomato against *Pseudomonas syringae*", the authors focused on tomato-*Pseudomonas syringae* interaction, aiming to unravel the ETI landscape in tomatoes. They used a comprehensive collection of *P. syringae* type III effectors called PsyTEC to investigate effector-triggered immunity (ETI) in various tomato varieties and cherry tomatoes. They found that tomatoes have a more limited ETI landscape compared to the model plant *Arabidopsis thaliana*. Only six effector families in *P. syringae* were capable of eliciting strong ETI responses in tomatoes. They also investigated whether ETI could be used to prime tomato plants' immunity against future infections. They discovered that ETI elicitors effectively protected tomato plants against *P. syringae* infection when delivered by a non-virulent strain either before or simultaneously with a virulent strain. In summary, this research provides valuable insights into the ETI landscape of tomato, revealing its distinct nature compared to *Arabidopsis thaliana*. Additionally, it demonstrates the potential of using ETI as a biocontrol strategy to enhance the protection of tomato crops against *P. syringae* infections. I think this research provides important and novel insights for the research field of plant immunity. However, the manuscript can be improved by the revision based on the comments below.

Major points

Line 421-425: In considering the potential utilization of ETI as a biocontrol strategy, it is essential to take into account the associated costs. ETI activation often leads to strong immune responses, culminating in a hypersensitive response (HR) and severe growth stunting in plants. Consequently, the concentration of the ETI-eliciting strain becomes a critical factor in achieving efficient pathogen protection while minimizing adverse effects on crop yield. It would be informative to investigate the concentration dependency of the ETI-eliciting strain on both pathogen protection and host plant growth.

Line 421-425: It is worth noting that tomatoes, which originated in the Andes Mountains and have spread worldwide relatively recently through human cultivation, may face unique challenges in dealing with *Pseudomonas* species as potential pathogens. It is conceivable that *Pseudomonas* species could be relatively new threats to tomatoes, and tomatoes might not have had sufficient time to evolve a comprehensive set of NLRs for recognizing them. In contrast, the interaction between *Arabidopsis* and *Pseudomonas* might have deeper historical roots, possibly extending further back in time. Consequently, *Arabidopsis* could have a more extensive repertoire of NLRs as a result of a longer co-evolutionary history with *Pseudomonas*. To better understand this dynamic, it would be intriguing to delve into the history of tomatoes, including their domestication timeline and their interactions with *Pseudomonas* species. Additionally, comparing the abundance and diversity of *Pseudomonas* species in the Andes Mountains, the tomato's place of origin, with other regions could provide valuable insights. Such a comparative analysis might shed light on whether the unique evolutionary history of tomatoes in the Andes has influenced their immune responses and NLR repertoire in response to *Pseudomonas* infections. Importantly, this perspective could help explain why wild tomatoes may not exhibit a wider recognition range of *Pseudomonas* effectors compared to cultivated tomatoes, highlighting the complex interplay between evolution, pathogens, and plant immune responses.

Line 418-420: There has been work on characterizing the effects of ETI-preactivation (or priming) leading to enhanced activation of PTI, and that PTI is required for ETI-induced resistance against Pst DC3000. Simultaneous activation of ETI and DC3000 infection leads to more robust immune responses (or less immunosuppression of immune responses by DC3000) in *Arabidopsis thaliana*. I believe the authors should mention these works in their discussion (<https://doi.org/10.1038/s41586-021-03315-7> and <https://doi.org/10.1038/s41586-021-03316-6>).

Line 95-96: 'the use of ETI as a targeted immunostimulant for priming has yet to be established'. This is not true. It has been shown that the pre-activation of ETI (using an inducible effector expression system) can prime *Arabidopsis thaliana* against DC3000 infection. Please cite the following work: <https://doi.org/10.1093/jxb/erz571>

Minor points

Line 26: "eliciting strong", do you mean strong immunity?

It would help the readers to navigate the article by stating exactly which figure they should be looking at (for example, Fig 4b-c instead of just Fig 4).

Reviewer #2 (Remarks to the Author):

Lonjon et al report on a large-scale analysis of possible effector recognition events leading to immunity in tomato (cultivated *S. lycopersicum*). Here, the authors leverage the previously well-built collection/library of *Pseudomonas syringae* effectors to investigate the potential of the tomato immune system to detect effector from this important bacterial pathogen. The *P. syringae* effector compendium (PsyTEC) was tested in *Arabidopsis* revealing the wide potential for effector-triggered immunity (ETI) in this model host. In the present study, using a similar high-throughput method to investigate ETI (evaluation of disease symptoms via automatized measurement of decline in greenness), the authors describe a narrower "ETI landscape" where only 6 (including 5 yet unknown) families of *P. syringae* effectors elicit ETI in tomato.

Based on their assessment of the conservation of this ETI profile across tens of cultivars and two closely related species (*S. arcanum* and *S. pimpinellifolium*), the authors thoroughly discuss the difference observed between tomato and *Arabidopsis*, including a possible loss during breeding, the presence of ETI-suppressing effectors in the strains used for the screen and the limitations due to the screening method.

In combination with their evaluation of effector family conservation across *P. syringae* species complex, the authors provide support and rationale for developing a new ETI-based protection methodology. Most of these currently rely on the identification/cloning/transfer of resistance loci mediating the effector recognition. New sources of resistance to bacterial speck disease are indeed identified in this work, although they remain to be characterized. More interestingly, the authors suggest and bring evidence that treatment (through spray-inoculation) with a strain carrying an ETI-eliciting effector prior to or concomitant with pathogenic strain infection protects tomato from bacterial speck disease.

The authors use a powerful approach to interrogate the immune potential of an important crop. Appropriate methods are used, and their limitations clearly stated. The findings from the screen are validated through alternative approaches (i.e. the decline in greenness as a proxy for disease symptoms evaluation is confirmed by bacterial enumeration in planta and ability of the identified effectors to trigger the hypersensitive response when delivered by the effectorless strain D36E). Some effector features (i.e. catalytic activity) required for recognition are further investigated and the possible nature of the resistance loci discussed. I commend the authors' efforts to provide representative raw data for the screen in different accessions. The manuscript is dense as it covers several levels of investigation (from an evolutionary perspective on both plant and pathogen to the molecular features of the recognition events). Nonetheless, this report is timely and of importance for a broad readership.

A few suggestions for improvement:

Table 1: add footnote for nRecPD (indication of HGT)

Table 2: add footnote for frequency parameters (e.g. number of strains examined)

Table 3: change "stains" to strains (in the second column)

Fig. 1 title: tomato ETI response to PsyTEC alleles (to replace "tomato response of PsyTEC")

Fig. 1: axes title? What is the variability of PtoDC3000(EV) growth? Formula for data normalization?

Fig. 7a: change "not expresses" to not expressed

L26: eliciting strong what?

L113: 21 families for ETI in Arabido, 19 families mentioned on L79

L137: change PtoDC000 to PtoDC3000

L138: add "Glamour" or "tomato" before plants

L143: how is the disease score distribution analyzed?
L154/252: add reference to Laflamme's study
L156: remove "." after "variation in. ETI"
L178: "high throughput screen", is it carried on 10-day-old tomato seedlings?
L218: revise the vague subheading – effector features required for ETI elicitation?
L288: quantification of "high degree" for horizontal gene transfer for HopAR1?
L342: reference for lack of RPS5 homolog in tomato?
L354: citing Fig. S12 here rather than Fig. 5
L414: not really/only "priming", change subheading in result section?
L420: "spatial structure in local acquired resistance". It would be worth to briefly develop this to clarify this point of discussion.
L433: 1M MgSO₄ is the final concentration in the medium? Effector expression
L434-5, L463, L467: add space between number and unit.
L439: references for PtoD36E and Pto delta avrPto delta avrPtoB strains?
L458: correct "umol/m2s".
L459: spray inoculated or sprayed with bacterial suspensions.
L467: bacterial suspension not solution
L471: change to bacteria counts? (instead of "concentrations")
L474: infiltrated into what?
L480: change incubated to incubating.
L487: add detection method/reagents.
Fig. S13 legend: lines 958-961 belong to the result section.

Please find below our responses to all comments from the two reviewers of our manuscript entitled “**The Effector-Triggered Immunity Landscape of Tomato against *Pseudomonas syringae***”. We thank the reviewers for their positive feedback and constructive recommendations. In response, we have addressed all concerns below and edited the manuscript accordingly. We have also added an author Yan Lai, who performed the requested experiments and finalized the manuscript revisions.

Reviewer #1 (Remarks to the Author):

Major points

Line 421-425: In considering the potential utilization of ETI as a biocontrol strategy, it is essential to take into account the associated costs. ETI activation often leads to strong immune responses, culminating in a hypersensitive response (HR) and severe growth stunting in plants. Consequently, the concentration of the ETI-eliciting strain becomes a critical factor in achieving efficient pathogen protection while minimizing adverse effects on crop yield. It would be informative to investigate the concentration dependency of the ETI-eliciting strain on both pathogen protection and host plant growth.

We appreciate the reviewer's suggestion to investigate the concentration dependency of the ETI-eliciting strain, and we have conducted experiments to address this aspect. We added in the result section and created figure S14 and have added the following statements to the text:

“In order to investigate the dose dependency of the ETI-induced protection against PtoDC3000, we conducted co-inoculation experiments with varying concentrations of D36E::HopAB1n (OD₆₀₀ 1.00, 0.33, 0.10, 0.033, and 0.01). Our results revealed that ETI-induced protection was dose-dependent. A robust protective effect was consistently observed against PtoDC3000 when D36E::HopAB1n was inoculated at the higher concentrations of OD₆₀₀ 1.0 and 0.33 (Fig. S14a), whereas no significant protection was observed at the lower concentrations of 0.1, 0.01 and 0.033 (Fig. S14a).”

As requested, we have also assessed the fitness cost of ETI-induction using plant fresh and dry-weight as a fitness proxy. These results are presented in Fig S14b,c and the following was added to the text:

“To assess any potential fitness costs associated with D36E-induced ETI, we sprayed 4-week old tomato plants with D36E::HopAB1n at OD₆₀₀ 1.00, 0.33, 0.10, 0.033, and 0.01 and measured the fresh and dry weight of the tomato plants 3 weeks post-inoculation. Our results indicate that there is no growth deficit associated with D36E::HopAB1n infection at any concentration tested as the fresh and dry weights of inoculated plants were comparable to control plants sprayed with MgSO₄ (Fig. S14b,c).”

Line 421-425: It is worth noting that tomatoes, which originated in the Andes Mountains and have spread worldwide relatively recently through human cultivation, may face unique challenges in dealing with *Pseudomonas* species as potential pathogens. It is conceivable that *Pseudomonas* species could be

relatively new threats to tomatoes, and tomatoes might not have had sufficient time to evolve a comprehensive set of NLRs for recognizing them. In contrast, the interaction between *Arabidopsis* and *Pseudomonas* might have deeper historical roots, possibly extending further back in time. Consequently, *Arabidopsis* could have a more extensive repertoire of NLRs as a result of a longer co-evolutionary history with *Pseudomonas*. To better understand this dynamic, it would be intriguing to delve into the history of tomatoes, including their domestication timeline and their interactions with *Pseudomonas* species. Additionally, comparing the abundance and diversity of *Pseudomonas* species in the Andes Mountains, the tomato's place of origin, with other regions could provide valuable insights. Such a comparative analysis might shed light on whether the unique evolutionary history of tomatoes in the Andes has influenced their immune responses and NLR repertoire in response to *Pseudomonas* infections. Importantly, this perspective could help explain why wild tomatoes may not exhibit a wider recognition range of *Pseudomonas* effectors compared to cultivated tomatoes, highlighting the complex interplay between evolution, pathogens, and plant immune responses.

We thank the reviewer for these thoughtful insights that will make for very interesting follow-up studies. As a teaser we have added the following to the Discussion:

“It is possible that tomatoes have a shorter co-evolutionary history with *P. syringae* than *Arabidopsis* resulting in a more limited resistance profile against this pathogen. Probing the diversity of *P. syringae* species in the Andes Mountains, where tomato originated, relative to areas of domestication may shed light on the evolutionary pressures that have shaped the current ETI landscape of tomato (Peralta, .E. & D.M. Spooner. 2000 ; PMID:31912142).”

Line 418-420: There has been work on characterizing the effects of ETI-preactivation (or priming) leading to enhanced activation of PTI, and that PTI is required for ETI-induced resistance against Pst DC3000. Simultaneous activation of ETI and DC3000 infection leads to more robust immune responses (or less immunosuppression of immune responses by DC3000) in *Arabidopsis thaliana*. I believe the authors should mention these works in their discussion (<https://doi.org/10.1038/s41586-021-03315-7> and <https://doi.org/10.1038/s41586-021-03316-6>).

We again thank the Reviewer for the thoughtful comment and have added the following to the Discussion: “Furthermore, it is possible that ETI-mediated protection involves the Pattern-Triggered Immune (PTI) response, since these two branches of the plant immune system have been demonstrated to mutually potentiate one another (Ngou et al. 2021, Yuan et al., 2021; PMID: 33692545; PMID: 33692546).”

Line 95-96: 'the use of ETI as a targeted immunostimulant for priming has yet to be established'. This is not true. It has been shown that the pre-activation of ETI (using an inducible effector expression system) can prime *Arabidopsis thaliana* against DC3000 infection. Please cite the following work: <https://doi.org/10.1093/jxb/erz571>

Thank you, we have included this reference to the Introduction:

“It has been demonstrated that pre-activation of ETI, utilizing an inducible effector expression system, can enhance priming against PtoDC3000 infection in *A. thaliana* (Ngou et al., 2020). However the effectiveness of using microbially-induced ETI as a targeted immunostimulant for priming in tomato has not yet been established.”

Minor points

Line 26: “eliciting strong”, do you mean strong immunity?

Corrected

It would help the readers to navigate the article by stating exactly which figure they should be looking at (for example, Fig 4b-c instead of just Fig 4).

Corrected

Reviewer #2 (Remarks to the Author):

A few suggestions for improvement:

Table 1: add footnote for nRecPD (indication of HGT)

We have added the footnote

Table 2: add footnote for frequency parameters (e.g. number of strains examined)

We have added a footnote

Table 3: change “stains” to strains (in the second column)

Corrected

Fig. 1 title: tomato ETI response to PsyTEC alleles (to replace “tomato response of PsyTEC”)

Corrected

Fig. 1: axes title? What is the variability of PtoDC3000(EV) growth? Formula for data normalization?

We have clarified this in the legend of Figure 1 : “Data represent the normalized results of twelve experiments. The minimum growth observed for the EV strain was 6.72 log CFU/cm², the maximum was 7.32 log CFU/cm², with an average of 7.12 log CFU/cm². For normalization within a single experiment, the growth of a particular replicate was divided by the average growth of the empty vector strain within that specific experiment.”

Fig. 7a: change “not expresses” to not expressed

Corrected

L26: eliciting strong what?

We changed to “electing a strong immunity”

L113: 21 families for ETI in Arabido, 19 families mentioned on L79

19 families were identified when the PtoDC3000 strains was used for screening, 2 more families were identified in the strain PmaES4326 (see references Laflamme et al., 2020 and Martel et al., 2022). This has been clarified in the Introduction:

“Systematic analyses of the ETI landscape of the model plant *A. thaliana* ecotype Col-0 against *P. syringae* identified 69 alleles from 21 families that can trigger ETI^{23, 24}.”

L137: change PtoDC000 to PtoDC3000

Changed

L138: add “Glamour” or “tomato” before plants

We added tomato

L143: how is the disease score distribution analyzed?

This has been clarified in the legend of Figure 2 “Distribution of disease scores (% green decline values) from the tomato var. Glamour PsyTEC screen (this study) (orange bars) and of the disease score values from the *A. thaliana* Col-0 PsyTEC screen²³ (green bars). The y-axis represents the number of effectors in each category when screened for ETI on the corresponding plant species and is presented as a log-scale.”

L154/252: add reference to Laflamme’s study

We added the reference.

L156: remove “.” after “variation in. ETI”

Removed

L178: “high throughput screen”, is it carried on 10-day-old tomato seedlings?

We modified: “We next comprehensively screened PsyTEC for ETI in the two wild tomato species using four week-old seedlings (Fig.3b,c S5, S6, Table 1).”

L218: revise the vague subheading – effector features required for ETI elicitation?

We modified the subheading to read: “Effector residues required for ETI and Hypersensitive Response elicitation”

L288: quantification of “high degree” for horizontal gene transfer for HopAR1?

This is quantified using a RecPD value as published in Bundalovic-Torma C, Desveaux D, Guttman DS. RecPD: A Recombination-aware measure of phylogenetic diversity. *PLoS Comput Biol* **18**, e1009899 (2022). “nRecPD values range from 0.0 to 1.0, with the former reflecting genes evolving primarily through horizontal evolutionary process, and the latter reflecting genes evolving primarily through vertical descent⁴¹. This is clarified in the text : “HopAR1g, HopAR1b, HopAR1a, HopAA1i, and HopBC1b, exhibit low nRecPD values (<0.3), indicating a high degree of horizontal transfer among *P. syringae* isolates.”

L342: reference for lack of RPS5 homolog in tomato?

We added the reference (PMID: 24885638)

L414: not really/only “priming”, change subheading in result section?

We changed to “ETI protects tomatoes against concomitant or subsequent *P. syringae* infection”

L420: “spatial structure in local acquired resistance”. It would be worth to briefly develop this to clarify this point of discussion.

We added “This finding may be due to spatial ETI structure described as localized acquired resistance⁵⁸, where the ETI response occurs in a 2 mm area surrounding cells in contact with the elicitor⁵⁸. This spatial structure implies that the primary protective impact of ETI may be localized to the initial infection site.”

L433: 1M MgSO₄ is the final concentration in the medium? Effector expression
Corrected

L434-5, L463, L467: add space between number and unit.
Corrected

L439: references for PtoD36E and Pto delta avrPto delta avrPtoB strains?
We added the references.

L458: correct “umol/m²s”.
Corrected

L459: spray inoculated or sprayed with bacterial suspensions.
We added “bacterial suspension”

L467: bacterial suspension not solution
Corrected

L471: change to bacteria counts? (instead of “concentrations”)
Changed

L474: infiltrated into what?
We added “tomato leaves”

L480: change incubated to incubating.
Modified

L487: add detection method/reagents.
We added “Signals were detected using Enhanced chemiluminescence (ECL) Western Blotting Substrates (BioRad).”

Fig. S13 legend: lines 958-961 belong to the result section.
This was removed from the Figure Legend since it was redundant with the Results section.

Reviewer #1 (Remarks to the Author):

The authors have addressed our previous feedback in the revised manuscript. I am satisfied with the responses provided and find the paper to be more interesting. Congratulations on your excellent work!

Reviewer #2 (Remarks to the Author):

The authors have satisfactorily addressed all the points raised by the reviewers. Although worthy of a dedicated follow-up study, the tomato/*P. syringae* co-evolution in the Andes is now discussed. The current report provides an excellent frame for further evolutionary considerations and deliver precious knowledge for developing field resistance strategies.